# Hypoxia Downregulates LPP3 and Promotes the Spatial Segregation of ATX and LPP1 During Cancer Cell Invasion

**DOI:** 10.3390/cancers11091403

**Published:** 2019-09-19

**Authors:** Kelly Harper, Karine Brochu-Gaudreau, Caroline Saucier, Claire M. Dubois

**Affiliations:** Department of Immunology and Cell Biology, Faculty of Medicine and Health Sciences, Université de Sherbrooke, Sherbrooke, Québec, QC J1H 5N4, Canada; kelly.harper@usherbrooke.ca (K.H.); Karine.brochu-gaudreau@usherbrooke.ca (K.B.-G.); caroline.saucier@usherbrooke.ca (C.S.)

**Keywords:** hypoxia, autotaxin, ATX, LPA, lipid phosphate phosphatases, LPP1, LPP3, cell invasion, invadopodia

## Abstract

Hypoxia is a common characteristic of advanced solid tumors and a potent driver of tumor invasion and metastasis. Recent evidence suggests the involvement of autotaxin (ATX) and lysophosphatidic acid receptors (LPARs) in cancer cell invasion promoted by the hypoxic tumor microenvironment; however, the transcriptional and/or spatiotemporal control of this process remain unexplored. Herein, we investigated whether hypoxia promotes cell invasion by affecting the main enzymes involved in its production (ATX) and degradation (lipid phosphate phosphatases, LPP1 and LPP3). We report that hypoxia not only modulates the expression levels of lysophosphatidic acid (LPA) regulatory enzymes but also induces their significant spatial segregation in a variety of cancers. While LPP3 expression was downregulated by hypoxia, ATX and LPP1 were asymmetrically redistributed to the leading edge and to the trailing edge, respectively. This was associated with the opposing roles of ATX and LPPs in cell invasion. The regulated expression and compartmentalization of these enzymes of opposing function can provide an effective way to control the generation of an LPA gradient that drives cellular invasion and migration in the hypoxic zones of tumors.

## 1. Introduction

Lysophosphatidic acid (LPA) is emerging as a critical oncogenic mediator regulating a variety of cellular processes implicated in tumorigenesis including cellular proliferation, cell motility, invasion and metastasis of a broad diversity of cancer cell types [1,2]. These effects of LPA are mediated by its signaling through specific G protein-coupled receptors (GPCRs), the LPA receptors (LPAR1–6), which activate a multitude of downstream responses associated with cytoskeletal remodeling, and the activation of kinases, integrins and matrix metalloproteinases [3,4,5]. Unsurprisingly, LPARs also promote the malignant progression of a variety of cancers including pancreatic, colon, liver, breast, endometrial and ovarian cancers [6,7,8,9,10,11]. Further highlighting the vital role of LPA signaling as a driver of tumor progression, overexpression of LPAR1-3 alone increases mammary tumorigenesis, invasion and metastasis in a mouse mammary tumor virus (MMTV) model [12].

High LPA levels are found in malignant ascites from ovarian and pancreatic cancer patients, and such increases represent an adverse prognostic factor [6,13,14]. Also, fast migrating melanoma cells use a self-generated LPA gradient to drive cell invasion [15]. Two groups of enzymes tightly control LPA levels by inducing its production and degradation. 

LPA production is mainly regulated by autotaxin (ATX), a secreted LysoPLD enzyme present in the cell supernatant of various malignant cell lines such as melanoma, glioblastoma and breast cancer [16,17,18]. ATX expression is upregulated in a wide range of malignancies including breast, lung, colon, ovarian, stomach, thyroid and brain cancers, correlating with the invasive potential of these cancer cells [18,19,20,21]. ATX localization can also be regulated as it binds to integrins or heparin sulfates on the cell surface, which may result in localized production of LPA close to LPA receptors [22,23]. Cellular characteristics associated with tumor aggressiveness, including cell proliferation, cell survival, cell motility, invasion, angiogenesis, resistance to treatment and metastasis, are all augmented by ATX, and depend on its capacity to produce LPA [24,25].

Opposing these effects of ATX are the major LPA degrading enzymes, lipid phosphate phosphatases (LPPs). LPPs are transmembrane proteins with extracellular catalytic domains. The LPP isoforms LPP1 and LPP3 are particularly implicated in the rapid degradation of extracellular LPA, resulting in the short half-life of this lipid mediator [26]. LPP1 knockout mice have an increase in the plasma level of LPA and a four-fold increase in the half-life of intravenously injected LPA, while extracellular LPA levels are significantly augmented in LPP3 knockout fibroblasts [27,28]. Interestingly, the expression levels of LPP1 and LPP3 are downregulated in some cancer types such as breast, lung and ovarian cancer [29]. In fact, low expression of LPP1 specifically is a contributing factor to the high levels of LPA found in ovarian cancer and the associated increases in proliferation, survival and migration [30,31]. The subcellular distribution of LPPs may also be regulated as LPP1 and LPP3 localize to distinct lipid raft domains, which has been proposed to result in spatial regulation of LPA signaling at the cellular level [32]. Therefore, increased production of extracellular LPA by ATX promotes tumorigenesis while reduced levels of LPA, due to the action of LPPs have a negative effect on tumor progression. Furthermore, spatial localization of these enzymes may result in specific microenvironments with elevated local production of LPA. However, the factors involved in shaping the differential expression or localization of these LPA regulatory components in the tumor microenvironment remain mostly unknown. 

An important tumor microenvironment factor driving tumor progression is hypoxia, a condition of low oxygen concentration, commonly arising in solid tumors due to their rapid proliferation limiting access to oxygen and nutrients [33,34]. Hypoxic tumors were found to be more aggressive, invasive and prone to recurrence [35,36,37]. At the molecular level, hypoxia activates a diverse array of transcription factors to profoundly affect cellular gene expression, as well as affecting cellular metabolism, acidification of the tumor microenvironment and trafficking of specific proteins involved in tumor progression [38,39,40]. Various studies investigating the cellular mechanisms responsible for hypoxia-induced cellular invasion found that it is associated with an increased production of invadopodia, which are degradative structures essential for cancer cell invasion and metastasis [41,42,43,44]. Additionally, a direct link between hypoxia and LPA signaling in cell invasion was established by our findings, indicating that hypoxic cancer cells rely on LPAR1 signaling through the phosphatidylinositol 3-kinase/Protein Kinase B PI3K/Akt pathway for invadopodia production and metastasis [45]. Due to their major roles in driving tumor progression, the interplay between hypoxia and the LPA axis warrants further exploration. 

Herein, we investigated whether the hypoxic tumor microenvironment stimulates cell invasion by affecting LPA regulatory enzymes. We found that hypoxia not only significantly diminishes the expression levels of LPP3 but also induces the spatial segregation of ATX and LPP1 at the plasma membrane. Such seclusion of ATX apart from LPP1, coupled with reduced LPP3 levels may result in uncontrolled LPA production at strategic cellular locations to drive cell invasion.

## 2. Results

### 2.1. Hypoxia Induces ATX and Represses LPP Expression in Certain Cell Lines

Given that tumor progression is associated with both an increase in the concentration of LPA within the tumor microenvironment and the extent of hypoxic areas within tumors [6,13,35,46], we sought to determine the impact of the hypoxic tumor microenvironment on the main enzymes that regulate LPA levels through its production and metabolism. To address this question, we first investigated the hypoxic regulation of the major LPA producing enzyme, ATX in diverse cancer cell lines including HT1080 fibrosarcoma, U87 glioblastoma, and MDA-MB231 breast cancer. A significant increase in ATX (*ENPP2*) mRNA expression was observed following 8 or 16 hours of hypoxic (1% O_2_) stimulation in HT1080 cells (Figure 1A). In contrast, U87 or MDA-MB231 cells showed no significant modulation (Figure 1B,C) despite a significant increase in mRNA expression of *CAIX*, an intrinsic marker of hypoxia (Figure 1D–F). These changes in mRNA expression correlated with protein expression for ATX (Appendix A, Appendix A).

LPPs also play an important role in controlling LPA levels. Thus, we next investigated whether hypoxia modulates the expression of LPPs in cancer cell lines. Aside from a transient but significant inhibition of LPP1 gene expression in U87 cells, no significant modulation of LPP1 or LPP2 was observed in HT1080, U87 or MDA-MB231 cells (Figure 1G–L). In contrast, hypoxia caused a pronounced decrease in LPP3 mRNA expression (up to 40%) in all three cell lines tested (Figure 1M–O). Changes in mRNA expression correlated with protein expression for LPP3 (Appendix A, Appendix A). Thus, hypoxia increases gene expression of the LPA-producing enzyme ATX while decreasing the expression of LPA degrading enzymes LPP1 and LPP3 in certain cancer cell lines, two events previously reported to lead to higher levels of LPA [27,28,47]. 

To gain insight into the importance of these findings in cancer, ATX gene expression and that of each of the LPPs were correlated with a set of genes previously found to be regulated by hypoxia in various cancers and to be predictive of patients likely to benefit from hypoxia-modifying therapy [48,49]. Using TCGA datasets of fibroblastic sarcoma, glioblastoma and triple negative breast cancer patient cohorts, we observed no significant correlation between gene expression of ATX and that of most of the eight hypoxia-regulated genes in the fibroblastic sarcoma and glioblastoma cohorts, while there was an overall negative correlation in breast cancer patients (Figure 2A–C). Of interest, we identified a striking negative correlation between the expression of most genes of the hypoxia signature and that of LPP3 in all three cancer patient cohorts, suggesting an association between the hypoxic tumor microenvironment and low levels of LPP3 gene expression in cancer patients (Figure 2J–L). In contrast, except for LPP1 in the sarcoma cohort, LPP1 and LPP2 showed inconsistent negative correlations, with the eight hypoxia-regulated genes in all three cancer patient cohorts (Figure 2D–I). Collectively, these results indicate that among the main enzymes regulating LPA production and degradation, only LPP3 is consistently regulated by hypoxia in cancers. 

Next, we used the SurvExpress online tool to define whether the downregulation of LPP3 gene in hypoxia is associated with a poor prognosis by assessing two prognostic factors (overall survival and metastasis-free survival). Results indicate that LPP3 expression can significantly separate low- and high-overall mortality risk groups in sarcoma, glioblastoma and breast cancer patient cohorts (Figure 2M–O). Markedly, an increased disparity between low- and high- metastasis-free risk groups was found in sarcoma and breast cancer patients cohorts, the two groups for which metastasis data was available (Appendix A, Appendix A). Moreover, low LPP3 gene expression was significantly associated with high-risk groups for both overall survival and metastasis-free survival in all three cohorts studied (Appendix A, Appendix A), suggesting that the hypoxic downregulation of LPP3 is linked to a poor prognosis.

### 2.2. ATX and LPP1 or LPP3 Exert Opposite Effects on Cell Invasion 

We have previously shown that hypoxia, through the LPA-LPAR1 receptor signaling axis, mediated cell invasion through production of actin/cortactin-rich invadopodial structures [43,45,50], In this study, we assessed whether the LPA synthesis and degradation enzymes were involved in this event. Results indicated that ATX knockdown by shRNA abolished invadopodia production induced by hypoxia, suggesting an important role for ATX in hypoxia-induced invadopodia production (Figure 3A). ATX was found to have a similar role when cells were allowed to migrate through 3D collagen-containing gels (Figure 3B–C). The efficiency of ATX gene knockdown was confirmed by qPCR (Appendix A, Appendix A) and western blotting [50]. Furthermore, under hypoxic conditions, the addition of the product (LPA) but not the substrate lysophosphatidylcholine (LPC) of ATX restored invadopodia production in ATX knockdown cells to the same level as in control cells (Figure 3D). This result indicates that the effects of ATX on hypoxia-induced invadopodia production are likely due to its ability to produce LPA 

Because LPP3, and to a lesser extent LPP1 expression levels are downregulated in hypoxic cells, we sought to determine whether reducing their cellular expression can recapitulate the effect of hypoxia on invadopodia production. For these experiments, we compared the effects of hypoxic stimulation (1% O_2_) to that of shRNA targeting each of the three LPPs. Results showed that LPP1 or LPP3 knockdown induced a significant increase in the percentage of cells forming invadopodia, comparable to the increase induced by hypoxia (Figure 3E). In contrast, knockdown of LPP2 had no effect on invadopodia production (Figure 3E), which is consistent with the lack of inhibition of LPP2 gene expression in hypoxia (Figure 1J). Immunofluorescence images showed an increase in matrix degradation in cells knockdown for LPP1 or LPP3 compared with control cells or cells knockdown for LPP2 (Figure 3F). Knockdown of LPPs gene expression by targeted shRNA was confirmed by qPCR (Appendix A, Appendix A). These results indicate that LPP1 and LPP3 play a negative role in invadopodia production, most likely through their known ability to degrade LPA [26].

### 2.3. Hypoxia Induces Spatial Segregation of ATX and LPPs

Despite the essential role of ATX in hypoxia-induced invadopodia production, ATX expression levels were not affected in most cell lines tested except a somewhat late 8–16 h, hypoxic induction in HT0180 cells (Figure 1A–C). However, hypoxia is also known to mediate effects by altering the trafficking and subcellular localization of various proteins [38]. Therefore, we sought to determine whether hypoxia might modulate the subcellular localization of ATX. First, HT1080 cells were permeabilized and stained for ATX and actin. Strong ATX staining at the leading edge of cells was observed in hypoxia compared to a more diffuse staining under normoxic conditions (Figure 4A). Because secreted ATX can be recruited to the cell-surface [22], we also performed ATX staining in non-permeabilized cells. Co-staining of the cells with the lipophilic marker DiD and ATX showed localized ATX staining at the cell surface (Figure 4B), as further seen in the associated z-axis images. In hypoxic cells, a prominent cell-surface staining was detected at the leading edge compared to a more diffuse staining under normoxic conditions (Figure 4B). Similar results were observed in hypoxic MDA-MB231 and U87 cells (Appendix A, Appendix A). These findings indicate that the subcellular localization of ATX is altered in hypoxic cells resulting in a marked redistribution of the enzyme to the leading edge. 

The spatial segregation of proteins with opposing functions was found to be essential to control cell signaling outputs [48]. Given the changes in ATX localization in hypoxia, we asked whether this condition also affected the localization of LPPs, particularly in relation to ATX localization. Double immunofluorescence staining of LPP1 and ATX in non-permeabilized HT1080, MDA-MB231 or U87 cells showed overlapping staining in normoxic cells, while hypoxic cells displayed staining for each protein in distinct cellular localizations. In hypoxia, ATX was detected at the leading edge, while LPP1 staining was located at the trailing edge with no apparent overlap with ATX in the different cell lines (Figure 5A, Appendix A, Appendix A).

In order to quantify these observations, we calculated the percentage of co-localization of these enzymes under normoxic versus hypoxic conditions. Results show that the co-localization of ATX and LPP1 is significantly reduced under hypoxic conditions in the three cell lines tested (Figure 6A–C). We performed the same double immunofluorescence staining with LPP2 and LPP3 and observed no apparent change in localization, corresponding with a lack of significant modulation in the percentage of co-localization with ATX in HT1080, MDA-MB231 or U87 cells (Figure 5B–C, Appendix A, Appendix A, Appendix A). These intriguing results suggest a significant spatial segregation of ATX versus LPP1 in hypoxic cancer cells, which, coupled with the observed reduction in LPP3 expression could result in excessive LPA production towards the leading edge of hypoxic cancer cells.

Given that LPAR1 was previously shown to mediate the hypoxic effect on HT1080 cell invasion [45], we next investigated LPAR1 distribution in these cells as well as their potential co-localization with ATX. Results show that LPAR1 are uniformly distributed in cells incubated under normoxic or hypoxic conditions (Figure 7A). Co-localization of LPAR1 with ATX indicates no significant changes in the percentage of co-localization under hypoxic conditions (Figure 7B). Collectively, the results suggest that while ATX and LPP1 are found polarized in opposite directions, LPAR1 is not segregated with either ATX or LPP1.

### 2.4. β1 and β3 Integrins Are Implicated in Cell-Surface ATX Localization 

Secreted ATX binds to β1 or β3 integrins, localizing ATX to the surface of platelets or cells such as lymphocytes and breast cancer cells [22,51,52]. In addition, β1 and β3 integrins are dynamically relocalized to the leading edge of polarized migrating cells [53], suggesting their participation in the recruitment of ATX to this cell area under hypoxia. Results of indirect immunofluorescence indicate strong co-localization of β1 and β3 integrins with ATX at the leading edge of hypoxic HT1080 cells, indicating that ATX could likely be bound to integrins at this location (Figure 8A). To determine whether ATX requires binding to integrins for localization at the cell surface, cells were pre-incubated with integrin function blocking antibodies. Results show a significant reduction in the percentage of cells with high ATX staining in cells incubated with β1 (60% reduction), β3 (48% reduction), or β1 and β3 (77% reduction) function blocking antibodies, while the percent of cells with low levels of ATX staining showed non-significant changes (Figure 8B). These results indicate that β1 and β3 integrins contribute significantly to the relocalization of ATX to the leading edge of hypoxic cells. 

## 3. Discussion

In this study, we uncovered hypoxia as a differential modulator of ATX and LPPs expression levels and as a regulator of their spatial distribution on the plasma membrane. While LPP3 expression was down regulated by hypoxia, ATX and LPP1 were asymmetrically redistributed to the leading edge and to the trailing edge, respectively. In agreement with their opposing roles in producing and degrading LPA, ATX and LPPs were found to promote or inhibit invadopodia production, respectively. Such seclusion of ATX apart from LPP1 coupled with reduced LPP3 levels, may result in higher LPA levels and uncontrolled LPA production at specific subcellular regions to drive cell invasion. 

Increased levels of LPA have been found in malignant effusions from breast, lung, kidney, lymphoma, ovarian and pancreatic cancer patients compared to control groups [6,13,46]. Ascitic fluids from ovarian cancer patients are hypoxic and conditions of hypoxia have been shown to increase LPA levels in non-malignant pathologies [54,55,56], leading to the possibility that hypoxia could be partly responsible for the observed increases in LPA levels in cancers. Here we provide evidence that even though hypoxia enhances the expression of the LPA producing enzyme, ATX in fibrosarcoma HT1080 cells, this effect is cell-specific as no regulation was observed in glioblastoma or breast cancer cells. Moreover, TCGA data analysis of breast cancer, glioblastoma and sarcoma indicate a lack of consistent relationship between the expression of ATX and hypoxia-regulated genes. These findings suggest that ATX regulation at the gene level is not a general way by which hypoxia regulates LPA levels in cancers, and reinforce the possibility that enhanced autocrine ATX production in cancer cells is mostly attributed to copy number amplifications or increased translation [2].

Besides ATX, LPA levels can also be affected by the amount of LPA degrading enzymes. Despite the downregulation of LPP1 and LPP3 in many tumor cells, little is known about the way LPPs are regulated [57]. While LPP1 was inconsistently down regulated, hypoxia significantly decreased LPP3 mRNA levels in all cell lines tested. The analysis of public data involving patient cohorts confirmed the association between low LPP3 levels and hypoxia in sarcoma, glioblastoma and breast tumors. Furthermore, low levels of LPP3 were found to correlate with increased risk of mortality and metastasis. We have therefore identified hypoxia as an important negative regulator of LPP3 expression that could contribute to the high levels of LPA found in the tumor microenvironment and poor prognosis [6,13,46].

Our results uncovered ATX as an essential mediator of hypoxia-induced invadopodia production and cell migration in a 3D matrix, as ATX knockdown completely blocked these events. Furthermore, LPA, but not the ATX substrate LPC, was able to rescue ATX shRNA-mediated knockdown of invadopodia in hypoxic conditions, indicating that the effects of ATX are most likely due to its production of LPA and downstream signaling through LPARs. This is consistent with previous results indicating that LPA production and LPAR signaling were responsible for the effects of ATX on invadopodia in normoxic conditions [50]. In contrast, knockdown of the LPA-degrading enzymes, LPP1 and LPP3, resulted in an increase in invadopodia production, an effect mimicking hypoxic stimulation. The identification of the suppressive role of LPPs in invadopodia production and their downregulation by hypoxia makes them relevant targets to block the LPA signaling axis in invading cancer cells. Increasing low LPP1/LPP3 expression through gene overexpression has been shown to limit tumor progression [2] and interestingly, tetracyclines can increase the expression of LPP1, LPP2 and LPP3 through stabilization of the protein [58]. It was thereby proposed that tetracyclines or other potential inducers of LPP gene, or protein, expression, or stabilization could be used to inhibit LPA signaling [2]. Clearly, more information on the mechanism by which LPPs are downregulated by hypoxia could provide novel targets for the control of LPP expression levels. 

An intriguing result of this study was the remarkably high level of ATX staining observed at the leading edge of hypoxic cancer cells and the associated spatial segregation of the LPA-producing, ATX, and LPA-degrading, LPP1, enzymes. This effect was isoform-specific as co-localization of ATX and LPP2 or LPP3 was not regulated by hypoxia. LPA signaling is affected by the amount of LPA available to interact with receptors, meaning that increased localized production of LPA near LPA receptors could enhance LPAR signaling [2]. Since we have shown that LPAR1 was the predominant LPA receptor responsive to hypoxia for cell invasion, one obvious question was to determine whether this receptor was also relocalized to the leading edge. Our findings that LPAR1 was uniformly distributed throughout the cell in both normoxia and hypoxia is consistent with other studies showing that chemoattractant receptors were uniformly distributed on the cell surface, while signaling was restricted to the leading edge [59]. Such absence of LPAR1 relocalization further suggests that it is indeed the compartmentalization of ATX/LPP1 under hypoxia that promotes cell migration through uncontrolled LPA production towards the leading edge that activates LPA receptors in this spatially restricted cell area.

It is now recognized that migrating cells must acquire spatial and functional asymmetry between the leading edge and rear of the cell, and one approach to such compartmentalization utilizes lipid rafts [60,61]. Interestingly, LPP1 and LPP3 have been found to localize to distinct lipid raft domains, with an enrichment of LPP3 in caveolin-1-positive lipid rafts and LPP1 in GM1-positive lipid rafts [32]. While lipid rafts are found redistributed at the leading edge in many cell types [53,60], a study in T lymphocytes found asymmetric redistribution of GM1 positive rafts to the uropod and GM3 positive rafts to the leading edge during cell migration [62]. These findings point to the possibility that selective lipid raft relocalization would lead to LPP1 redistribution to the trailing edge of hypoxic cells and explain why LPP1 but not LPP3 was segregated from ATX in hypoxic cells.

Recently, β1 and β3 integrins were implicated in recruiting ATX to the leading edge of cancer cells promoting persistent directional migration [63]. In concordance with these studies, we found integrins to be implicated in ATX localization to the cell surface of hypoxic cancer cells. Such recruitment of integrin-bound ATX to the leading edge of hypoxic cells might be facilitated by the fact that hypoxia affects integrin recycling as it stimulated Rab11-dependent recycling of integrin α6β4 to the plasma membrane [64]. This effect of hypoxia was associated with increased invasion and migration by maintaining integrins at the leading edge of cells [65]. 

Cell invasion is a complex process that includes the formation of anterior protrusions at the leading edge and localized extracellular matrix (ECM) degradation mediated by invadopodia. Interestingly, β1 and β3 integrins are also required for the formation of mature degradation-competent invadopodia [66,67]. Therefore, the interaction of ATX with β1 and β3 integrins in hypoxia may recruit ATX to cell protrusions at the leading edge or sites of invadopodia formation where the enzyme can deliver LPA locally to receptors. These receptors can subsequently drive processes, which promote cytoskeletal rearrangements at the leading edge of migrating cells or at sites of invadopodia production that include PI3K activation [68,69,70,71]. This suggests that inhibition of the ATX-integrin interaction could be a selective approach to prevent localized LPA signaling that drives cell invasion in hypoxic cancer cells. In this regard, generation of a blocking peptide for this interaction could be envisioned since the crystal structure of ATX as well as the specific amino acids involved in ATX-integrin association have been determined [51]. Such a strategy has the potential to generate fewer side effects than global ATX inhibition. 

## 4. Materials and Methods 

### 4.1. Reagents 

1-oleoyl-sn-glycerol-3-phosphate 18:1 (LPA) sodium salt and 1-oleoyl-*sn*-glycero-3-phosphocholine 18:1 (LPC) were from Sigma-Aldrich (St. Louis, MO, USA). Plasmid ATX cDNA construct was a kind gift from Dr. Tim Clair (Center for Cancer Research, NCI, NIH, Bethesda, MD, USA). shRNA against ATX or non-targeted (ctr) shRNA was from SABiosciences (Frederick, MD, USA). Mission lentiviral shRNA targeting LPP1 (TRCN0000010720 and TRCN0000002579), LPP2 (TRCN0000002583 and TRCN0000002584), LPP3 (TRCN0000358710 and TRCN0000358709), or a scramble sequence, were from Sigma-Aldrich. ATX (Mouse), LPP1, LPP2 and LPP3 antibodies were from Abcam (Cambridge, UK). ATX (Rabbit) antibody was from Cayman Chemical (Ann Arbor, MI, USA), β1 integrin antibody (P4C10) was from Millipore Sigma (Etobicoke, ON, Canada), β3 integrin antibody was from Bio-Rad (Hercules, CA, USA) and LPAR1 antibody was from Novus Biologicals (Littletone, CO, USA). Tubulin antibody was from Sigma-Aldrich. Texas Red phalloidin, DAPI (4’,6-diamidino-2-phenylindole), the lipophilic tracer DiD, and all secondary fluorophore-coupled antibodies were from Invitrogen (Molecular Probes, Eugene, OR, USA). HRP-coupled secondary antibodies were from Cell Signaling technology (Danvers, MA, USA). Fibrillar collagen type I was from R&D Systems (Minneapolis, MN, USA).

### 4.2. TCGA RNAseq Data Analysis

Gene expression from publicly available TCGA Illumina HiSeq RNA-Seq (RSEM normalized) datasets was obtained through the TCGA Data Portal (http://cancergenome.nih.gov). The sarcoma cohort of 265 patients (SARC-TCGA, provisional) was filtered for fibroblastic sarcomas (*N* = 86; undifferentiated pleomorphic sarcomas, synovial sarcomas, myxosarcomas and desmoid/aggressive fibromatosis). The glioblastoma cohort (Glioblastoma Multiforme, TCGA Pan-Cancer Atlas study) was composed of 166 patients; and the basal breast cancer cohort (*N* = 171) was obtained by filtering the basal cancer subtype from the breast invasive carcinoma TCGA cohort of 1082 samples [72]. Pairwise Spearman correlation coefficients (*r*) were calculated between genes of the LPA producing/degrading pathway (*ENPP2, PLPP1, PLPP2* and *PLPP3*) and the hypoxia signature genes (*PDK1, PGAM1, LDHA, NDRG1, CDKN3, TUBB6*, *MIF* and *MRPS17*) [73] for each cancer patient cohort. Affymetrix gene expression results and associated overall survival and metastasis-free survival data from sarcoma (GSE21050, [74]), glioblastoma (GSE13041, [75]) and breast ([76]) cancer patient cohorts were used to evaluate the correlation between *PLPP3* gene expression and the determination of high- and low-risk patients using the publicly available online software SurvExpress (http://bioinformatica.mty.itesm.mx:8080/Biomatec/SurvivaXvalidator.jsp) [77]. 

### 4.3. Cell Culture and Transfection

HT1080 human fibrosarcoma, MDA-MB231 human breast cancer and U87 human glioblastoma cells were obtained from the American Type Culture Collection (ATCC, Rockville, MD, USA). Cells were cultured in minimal essential medium (MEM) (Wisent, St-Bruno, QC, Canada) supplemented with 10% FBS (Gibco BRL, Burlington, ON, Canada) and 40 µg/mL of gentamicin (Wisent Inc, St-Bruno, QC, Canada) in a humidified 95% air/5% CO_2_ incubator at 37 °C. For hypoxic stimulations, cells were cultured in an INVIVO2 400 hypoxic chamber (Ruskinn, Sanford, ME, USA) at 1% O_2_ and 5% CO_2_. For experiments involving stable transfections, with ATX, pcDNA3.1 or shRNA against ATX or non-targeted sequence, cells were seeded at a density of 1 × 10^5^ cells per well in a 6-well culture plate one day before transfection. Transfections were performed with the Fugene reagent (Roche Diagnostics, Mannheim, Germany), according to the manufacturer’s protocol. Stable transfectants were obtained by antibiotic selection, G418 (600 µg/mL; Gibco, Thermo Fisher Scientific, Waltham, MA, USA) for ATX and pcDNA3.1 transfections and Puromycin (2 µg/mL; Invivogen, San Diego, CA, USA) for all shRNA transfections. For lentiviral transductions, with LPP1, LPP2, LPP3 or scramble shRNA, cells were seeded at a density of 3 × 10^5^ cells per 10 cm^2^ Petri dish and infected with 1 mL of viral stock in 2 mL of optiMEM supplemented with 2 μL Polybrene (10 mg/mL; EMD Millipore, Etobicoke, ON, Canada). Viral particles were generated by transient transfection of 293T cells using a ViraPower lentiviral expression system (Invitrogen Thermo Fisher Scientific, Burlington, ON, Canada).

### 4.4. Real Time RT-PCR

Total RNA was isolated using the TRIzol (Invitrogen, Carlsbad, CA, USA) protocol as previously described [50] and 1 µg of RNA was reverse transcribed to complementary DNA (cDNA) using a QuantiTect reverse transcription kit (Qiagen, Mississauga, ON, Canada). cDNA was then analyzed by real time PCR using a hot start SYBR Green qPCR master mix (BiMake, Houston, TX, USA). The following primer pairs were selected for: ATX: (forward) 5’-TGAAACAGCACCTTCCCAAA-3’, (reverse) 5’-CCAAAGGTTTCCTTGCAACA-3’; LPP1: (forward) 5’-GTCGAGGGAATGCAGAAAGA-3’, (reverse) 5’-CCTTCATCCTGGCTTGAAGATA-3’; LPP2: (foward) 5’-CCTACCGTCCAGATACCATCA-3’, (reverse) 5’-GTTGAAGTCCGAGCGAGAATAG-3’; LPP3: (forward) 5’-CAAATCAGAAGGAGCCAGAGAA-3’, (reverse) 5’-CAGCAAGAGCAACTCCTACAA-3’; CAIX: (forward) 5’-CCTCAAGAACCCCAGAATAATGC-3’, (reverse) 5’-CCTCCATAGCGCCAATGACT-3’; and housekeeping gene RPLP0: (forward) 5’-GATTACACCTTCCCACTTGC-3’, (reverse) 5’-CCAAATCCCATATCCTCGTCCG-3’. 

Quantitative Real-Time PCR was performed on a Rotor-Gene 3000 (Corbett Research, Kirkland, QC, Canada). The cycling program was as follows: initial denaturation at 95 °C for 15 min, 35 amplification cycles with annealing T of 59 °C for 30 s and final extension at 72 °C for 30 s. Results were calculated as 2^ΔΔCT^.

### 4.5. Western Blotting

Cells were lysed in RIPA buffer and immunoblotting was performed as previously described [50]. Membranes were probed overnight with primary antibodies. The secondary antibody was a peroxidase-conjugated anti-rabbit or anti-mouse antibody, depending on the source of primary antibody used. Immunoblots were revealed using the Luminata^TM^ Western HRP Chemiluminescence substrate (Millipore, Etobicoke, ON, Canada).

### 4.6. Invadopodia Assay

Coverslips were prepared as previously described [50], using Oregon-Green^488^-conjugated gelatin (Invitrogen, Burlington, ON, Canada). Forty thousand cells were seeded on each coverslip and allowed to adhere. Following various incubation times as described within the figure legends, cells were fixed with 2% paraformaldehyde for 10 min at room temperature. Nuclei were stained with DAPI and F-actin was stained using Texas-Red-conjugated phalloidin. Cells were visualized and imaged by fluorescence microscopy using an Axioskop 2 phase-contrast/epifluorescence microscope (Carl Zeiss, Inc., Thornwood, NY, USA). Cells forming ECM-degrading invadopodia were identified based on cells with at least one F-actin-enriched area of matrix degradation (characterized by loss of green fluorescence). Three fields of 100 cells (magnification 40×) were counted per coverslip to quantify the percentage of cells forming ECM-degrading invadopodia. 

### 4.7. 3D Invasion Assay

Collagen type I 3D matrix was prepared as follows: Aliquots (50 µL) of agarose-containing 10% FBS were deposited in a 96 well culture plate. Aliquots (50 µL) of fibrillar collagen type I (R&D Systems, Minneapolis, MN, USA) were prepared following manufacturer’s instructions and layered on top of the agarose. Cells (2 × 10^4^/100 µL in serum-free MEM) were deposited on top of the collagen gel and incubated for 24 h. The cells were then labeled with CellTrace^TM^ calcein green AM (Invitrogen, Burlington, ON, Canada) 1 h prior to the end of incubation. Cells were washed with PBS and fixed with 3% glutaraldehyde for 30 min followed by confocal microscopy analysis using a FV1000 Olympus confocal microscope. Collagen matrix pellets were scanned along the Z-axis. Cells that had invaded the collagen were imaged and quantitated at each 5 µm layer within the gel.

### 4.8. Immunofluorescence

Forty thousand cells were seeded on non-fluorescent gelatin-coated coverslips and allowed to adhere for 30 min. Following stimulations, as indicated in figure legends, cells were fixed with 2% paraformaldehyde in PBS for 10 min at room temperature. Where indicated, cells were permeabilized with 0.05% saponine (Sigma-Aldrich, St. Louis, MO, USA) in PBS for 20 min and blocked with 2% BSA in PBS for 30 min. Then, cells were incubated with the appropriate primary antibodies for 2 h, and secondary antibodies for 1h or fluorescent phalloidin for 45 min, as indicated within the Figure legends. Images were taken with a FV1000 scanning confocal microscope (Olympus, Tokyo, Japan) coupled to an inverted microscope using a 63× oil immersion objective. For quantification of ATX/LPP or ATX/LPAR1 co-localization, cells were incubated with ATX and LPP1, LPP2, LPP3, or LPAR1 antibodies. The percentage of co-localization was calculated as previously described from serial optical sections of the whole cell [78].

### 4.9. Integrin Blocking Assay

Forty thousand cells were seeded on non-fluorescent gelatin-coated coverslips and allowed to adhere for 30 min. Cells were then incubated with β1 β3, or β1 + β3 integrin blocking antibodies, or control IgG mouse for 30 min prior to 3 h incubation in hypoxia (1% O_2_). Cells were then fixed with 2% paraformaldehyde in PBS for 10 min at room temperature and blocked with 2% BSA in PBS for 30 min. Then, cells were incubated with ATX antibody (Rb) for 2 h, and secondary antibody for 1h followed by DAPI for 5 min. Cells were visualized by fluorescence microscopy using a Zeiss Axioskop fluorescence microscope. Cells with strong or moderate green fluorescent ATX staining were counted. Cells that display an absence of ATX staining were counted as negative. Three fields of 100 cells (magnification 40×) were counted per coverslip to quantify the percentage of cells with ATX staining.

### 4.10. Statistical Analysis 

The GraphPad software was used for statistical analysis. Unless otherwise indicated, paired or unpaired Student’s t-test were used to assess statistical significance, which was set at a *p* value <0.05.

## 5. Conclusions

The role of hypoxia in cell invasion appears to require the tight control of LPA bioactivity in extent and subcellular space, events that would likely be crucial for directional ECM degradation and cell movement. Additional work will continue to elucidate how hypoxia modulates and segregates these important LPA regulatory enzymes to provide additional cues for the design of therapies targeting these important aspects of tumor progression. 

## Figures and Tables

**Figure 1 cancers-11-01403-f001:**
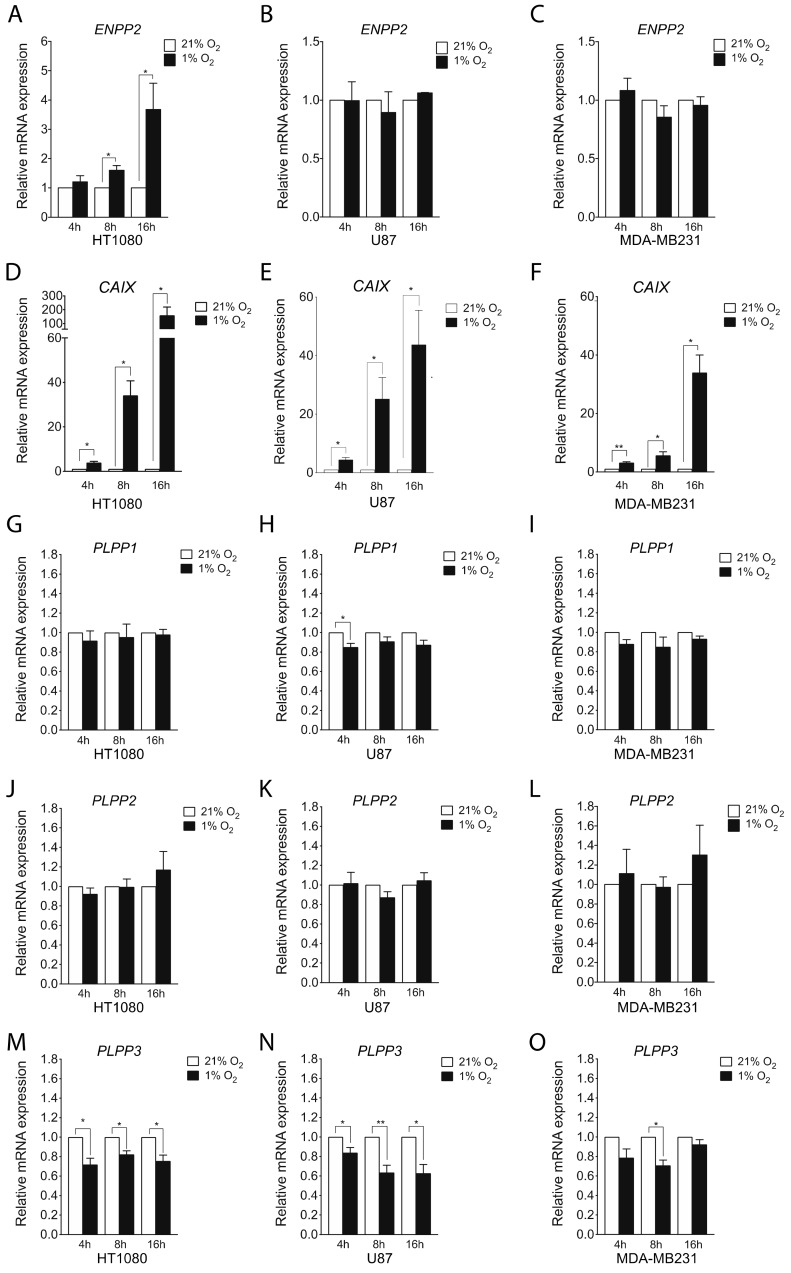
Effect of hypoxia on expression of autotaxin (ATX) and lipid phosphate phosphatases (LPP). (**A**–**J**) Cells were incubated under normoxic (21% O_2_) or hypoxic (1% O_2_) conditions for 4, 8 or 16 hours. mRNA expression of (**A**–**C**) *ENPP2* (autotaxin), (**D**–**F**) *CAIX* (carbonic anhydrase IX), (**G–I**) *PLPP1* (LPP1), (**J**–**L**) *PLPP2* (LPP2) or (**M**–**O**) *PLPP3* (LPP3) was evaluated by qPCR in (A, D, G, J, M) HT1080, (B, E, H, K, N) U87, or (C, F, I, L, O) MDA-MB231 cells. *RPLP0* was used to normalize the data. *N* ≥ 3. Bars represent the mean ± SEM (* *p* < 0.05, ** *p* < 0.01).

**Figure 2 cancers-11-01403-f002:**
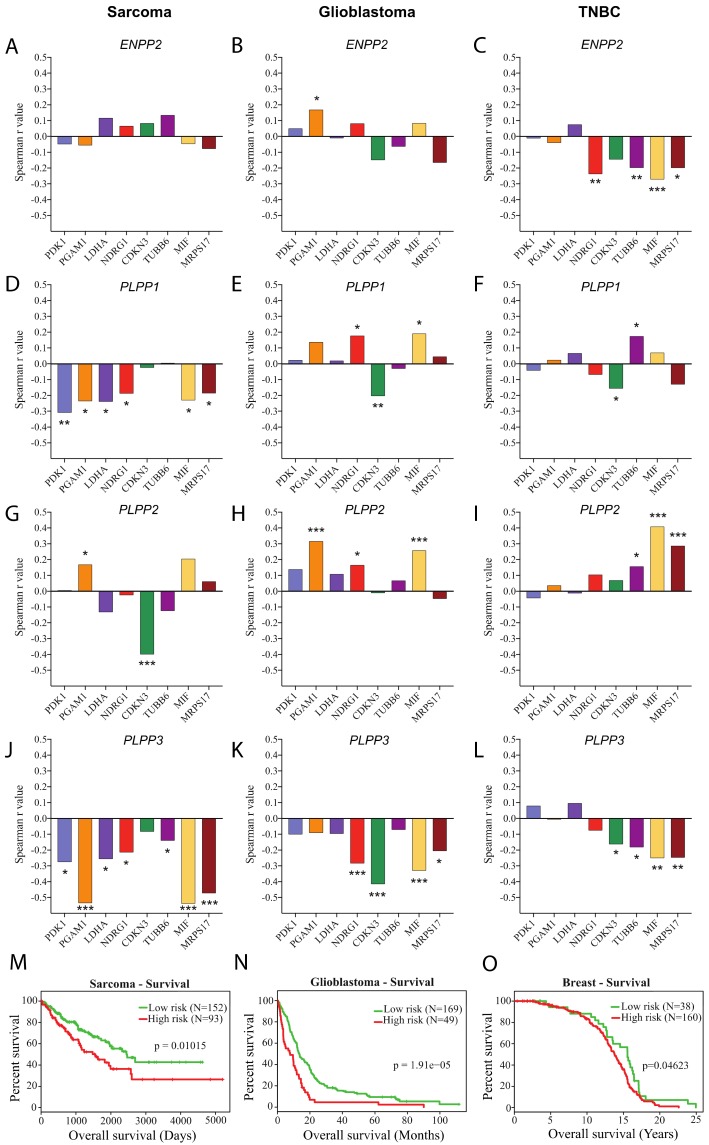
Correlation between ATX and LPP gene expression with a hypoxia gene signature and risk of mortality in patient cohorts. TCGA RNAseq data was used to measure Spearman r correlation coefficient of (**A**–**C**) *ENPP2* (autotaxin), (**D**–**F**) *PLPP1* (LPP1)*,* (**G**–**I**) *PLPP2* (LPP2) or (**J**–**L**) *PLPP3* (LPP3) RNA expression with hypoxia-induced genes in (**A**,**D**,**G**,**J**) fibroblastic sarcoma (*N* = 86), (**B**,**E**,**H**,**I**) glioblastoma (*N* = 166), or (**C**,**F**,**I**,**L**) basal breast cancer (*N* = 171) tumor tissue from patient cohorts. (* *p* < 0.05, ** *p* < 0.01, *** *p* < 0.001). (**M**–**O**) Kaplan-Meier plots obtained using the SurvExpress online software showing overall survival curves of high- and low-prognostic risk groups based on *PLPP3* expression in sarcoma (M), glioblastoma (N) and breast (O) cancer patients cohorts. Log-rank test *p*-values are presented.

**Figure 3 cancers-11-01403-f003:**
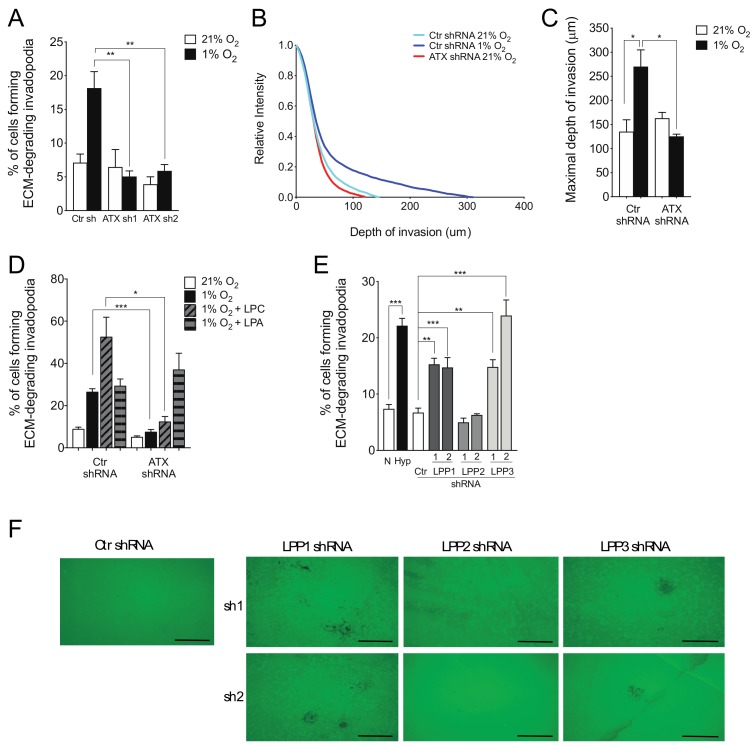
Role of ATX and LPPs in cancer cell invasion. (**A**) Cells transfected with non-targeting control (Ctr) or one of two ATX-targeted shRNA constructs (sh1 or sh2) were cultured for 10 h on fluorescently-labeled gelatin. The percentage of cells forming extracellular matrix (ECM)-degrading invadopodia is shown for cells cultured in normoxia (21% O_2_) or hypoxia (1% O_2_) *N* = 3. (**B**–**C**) Cells transfected with non-targeting (Ctr) or ATX-targeted shRNA were incubated on type I collagen in 3D invasion assays in normoxia (21% O_2_) or hypoxia (1% O_2_) for 24 h. (**B**) The relative intensity of cell staining according to depth of invasion is shown. (**C**) The maximal depth of invasion is shown for each condition. (**D**) Cells transfected with non-targeting control (Ctr) or ATX-targeted shRNA were incubated in normoxia (21% O_2_), hypoxia (1% O_2_), hypoxia with LPC 10 μM (1% O_2_ + LPC), or hypoxia with LPA 10 μM (1% O_2_ + LPA). The percentage of cells forming ECM-degrading invadopodia is shown, *N* = 3. (**E**,**F**) HT1080 cells incubated in normoxia (21% O_2_) or hypoxia (1% O_2_), or HT1080 cells transduced with non-targeting control (Ctr) or one of two LPP1-, LPP2-, or LPP3-targeted shRNA constructs incubated in normoxia (21% O_2_) were cultured for 10 h on fluorescently-labeled gelatin. (**E**) The percentage of cells forming ECM-degrading invadopodia and (**F**) representative images of matrix degradation are shown. *N* = 3. Bars represent the mean ± SEM (* *p* < 0.05, ** *p* < 0.01, *** *p* < 0.001). Scale bars, 50 μm.

**Figure 4 cancers-11-01403-f004:**
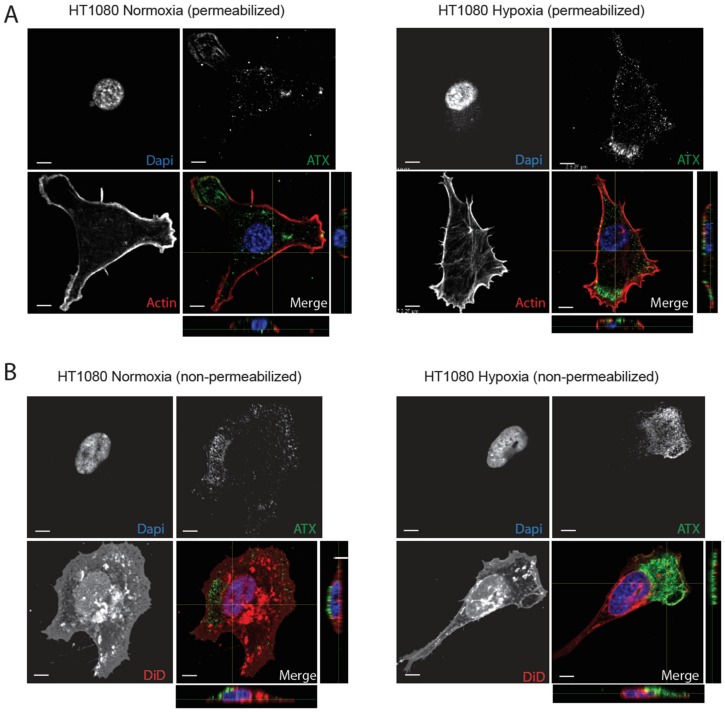
ATX localization in hypoxic cancer cells. (**A**–**B**) Representative immunofluorescence images of HT1080 cells cultured on non-fluorescent gelatin for 4 h in normoxia (21% O_2_) or hypoxia (1% O_2_) are shown. (**A**) Representative images of cells permeabilized and stained for ATX (green) or F-actin (red). Nuclei were stained with DAPI (blue). Magnification 60×, scale bars, 5 μm. (**B**) Representative images showing ATX (green) and the lipophilic carbocyanine DiD (red) staining in non-permeabilized cells. Nuclei were stained with DAPI (blue). Magnification 60×, scale bars, 5 μm.

**Figure 5 cancers-11-01403-f005:**
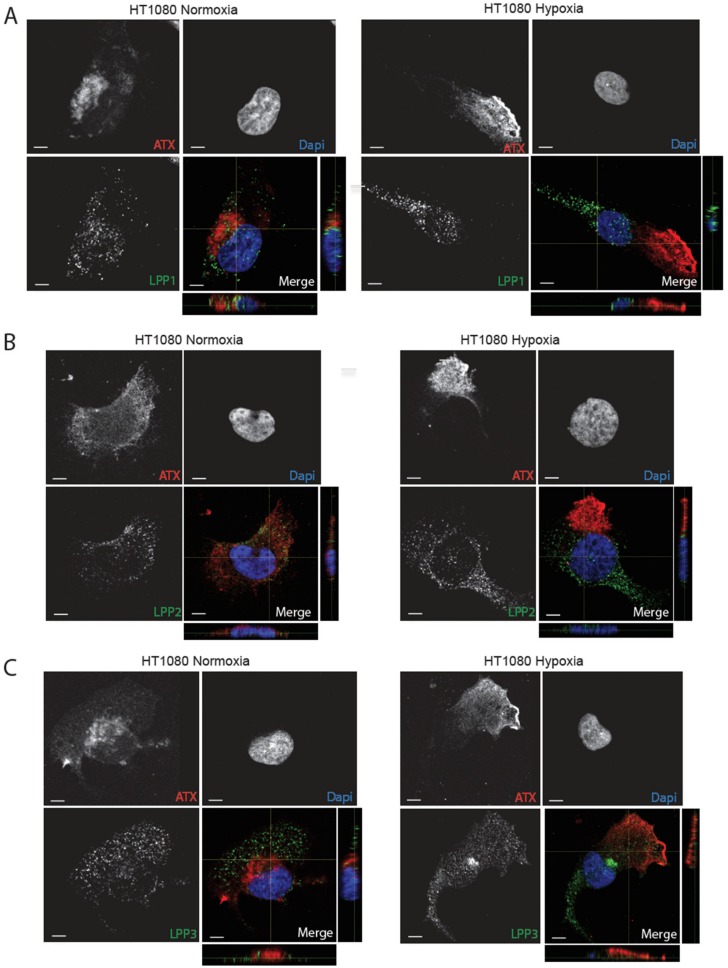
Localization of both ATX and LPPs in hypoxic cancer cells. (**A**–**C**) Representative immunofluorescence images of HT1080 cells cultured on non-fluorescent gelatin for 4 h in normoxia (21% O_2_) or hypoxia (1% O_2_) are shown. Cells were stained for (**A**) LPP1, (**B**) LPP2, or (**C**) LPP3 (green) and ATX (red). Nuclei were stained with DAPI (blue). Magnification 60×, scale bars, 5 μm.

**Figure 6 cancers-11-01403-f006:**
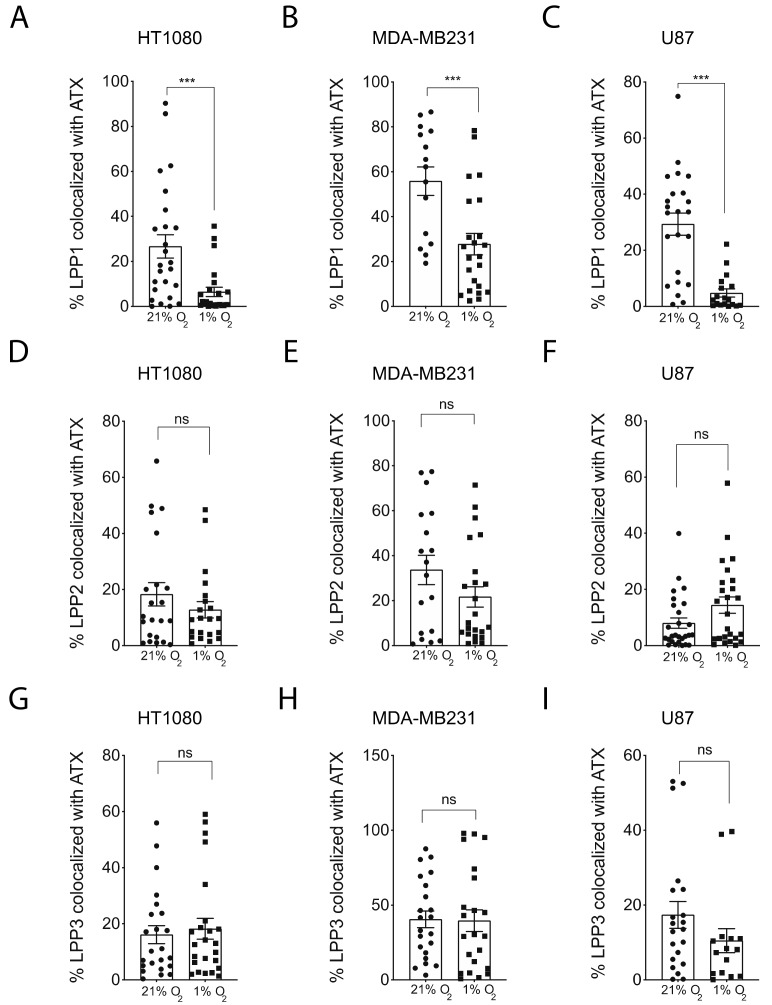
Quantification of the co-localization of ATX and LPPs under hypoxic conditions. Quantification of the percentage of (**A**–**C**) LPP1, (**D**–**F**) LPP2, or (**G**–**I**) LPP3 co-localized with ATX in (**A**,**D**,**G**) HT1080, (**B**,**E**,**H**) MDA-MB231 or (**C**,**F**,**I**) U87 cells cultured on non-fluorescent gelatin for 4 h in normoxia (21% O_2_) or hypoxia (1% O_2_). Bars represent the mean ± SEM (*** *p* < 0.001, ns = non-significant).

**Figure 7 cancers-11-01403-f007:**
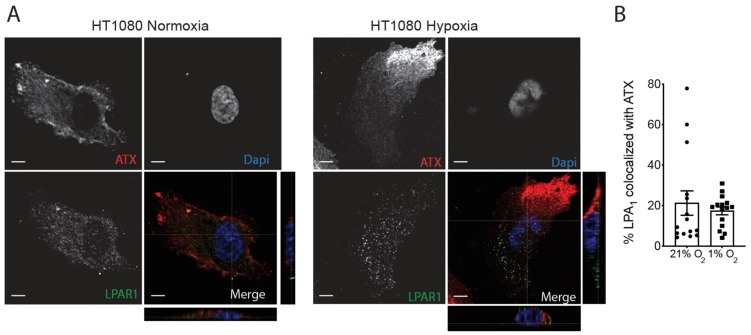
Localization of LPAR1 in hypoxic cancer cells. HT1080 cells were cultured on non-fluorescent gelatin for 4h in normoxia (21% O_2_) or hypoxia (1% O_2_). (**A**) Representative immunofluorescence images of cells stained for LPAR1 (green) and ATX (red). Nuclei were stained with DAPI (blue). Magnification 60×, scale bars, 5 μm. (**B**) Quantification of the percentage of LPAR1 colocalized with ATX in HT1080 cells. Bars represent the mean ± SEM.

**Figure 8 cancers-11-01403-f008:**
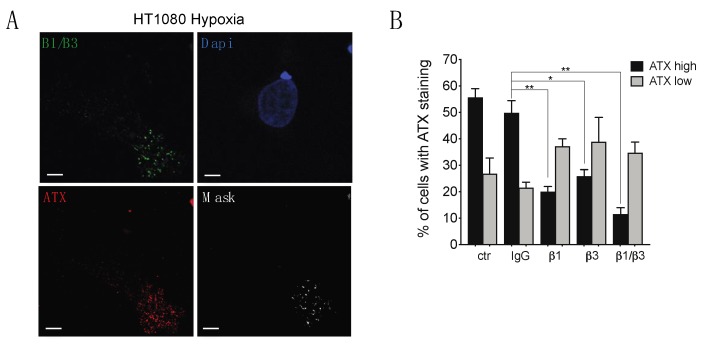
Evaluation of integrin contribution to the subcellular localization of ATX in hypoxic cancer cells. (**A**) Representative immunofluorescence images of HT1080 cells cultured on non-fluorescent gelatin for 4 h in hypoxia (1% O_2_) are shown. Cells were stained for ATX (red) and β1 and β3 integrins (green). Nuclei were stained with DAPI (blue). Mask overlay shows co-localization of ATX and integrins (white). Magnification 60×, scale bars, 5 μm. (**B**) HT1080 cells were plated on non-fluorescent gelatin and incubated with control (ctr) IgG, β1, β3 or β1 and β3 integrin blocking antibodies for 30 min followed by 3 h incubation under hypoxia (1% O_2_). Cells were then stained for ATX (green) and nuclei were stained with DAPI (blue). The percentage of cells with ATX staining (high/low) is shown. *N* = 3. Bars represent the mean ± SEM (* *p* < 0.05, ** *p* < 0.01).

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
