# Peer review of "Hypoxia Downregulates LPP3 and Promotes the Spatial Segregation of ATX and LPP1 During Cancer Cell Invasion"

_cancers, 2019, doi:10.3390/cancers11091403_

Round 1

Reviewer 1 Report

Reviewers comments for K. Harper et al:

Hypoxia Down regulates LPP3 and Promotes the 2 Spatial Segregation of ATX and LPP1 During Cancer 3 Cell Invasion

Overview:

Research presented in this paper has relevance to a broad range of clinical and basic researchers working in discovery-led translational cancer research. The manuscript aims to define the role that positive and negative regulators of LPA abundance may play in regulating metastasis under hypoxic conditions.

Although this manuscript is a logical (yet essential) progression from previous high-quality research (REF 44), it is important to acknowledge that novel data presented in this manuscript successfully resolves a fundamentally important question; Are changes in enzymes that regulate LPA abundance sufficient to enhance metastatic potential?  This manuscript clearly demonstrate that this is the case. As such, the significance of the paper lies in the fact that the authors demonstrate that reciprocal cancer-related changes in ATX and LPP activity (underlying mechanisms not yet defined) are dominant factors in hypoxic metastatic progression. Significantly, when these factors are perturbed there appears to be no other compensatory mechanism that maintain full metastatic potential. This observation highlights a logical focus for future therapeutic intervention in a broad range of solid tumours.

Despite the quality of the data presented in this manuscript, there are some surprising gaps in information. For example, do hypoxic changes in ATX/LPP3 mRNA levels lead to corresponding changes in protein expression, or activity? In this context, it is strange that comparative levels of protein expression are not shown under normoxic and hypoxic conditions.  Irrespective of the magnitude of any observed changes, this is an important piece of the jigsaw. If protein levels stay the same, functional modulation may be posttranslational. This is important to know, whatever the result, this should not detract from the value of current data.

The magnitude of change imposed by shRNA depletion may not be equivalent to hypoxia induced changes in gene expression. Therefore, it is important to define the extent to which expression of ATX and LPP proteins changed under hypoxic conditions?

Equally, as the authors state that the relative localisation of LPP proteins and LPA receptors may affect local LPA concentrations it would be informative to know the relative distribution of LPA receptors under normoxic and hypoxic conditions, are they localised with ATX or LPPs?

Finally, one of the most significant observations in this manuscript is the report that there is a significant reciprocal correlation between the expression of hypoxic genes and LPP3. However, we are not informed if decreased LPP3 expression correlates with stage specific changes or if reduced LPP3 levels correlate with poor prognosis/survival profile etc. This data should be available and would be informative wrt the potential therapeutic significance of these observations.

Responses to specific Journal criteria are detailed below:

1). Originality: This is a logical, yet valuable continuation of previous high-quality research. Nevertheless, the manuscript presents a significant amount of high-quality novel data, which extends mechanistic understanding of the functional consequences of hypoxic conditions in tumour progression and metastasis. Although there are some gaps in the data, which would significantly enhance the utility of the current manuscript, these are desirable rather than essential.

2). Significance: Data presented in this manuscript provides novel insight into the molecular mechanisms by which hypoxia may contribute to metastatic progression in a broad range of solid tumours. As such the manuscript will be directly relevant to may areas of cancer research, and therefore has the potential for high citation.

3). Quality of presentation: The manuscript is written to a very high standard, text and figures are excellent.

4). Scientific soundness: All experiments are appropriately designed and correctly controlled, there are no concerns wrt data quality.

5). Interest to readers: As stated above this paper addresses issues that are relevant to researchers working on a diverse spectrum of solid tumors. As such, the manuscript will have relevance to many areas of cancer research.

6). Overall merit is strong, although there are obvious gaps in the data.

Reciprocal changes in hypoxia induced mRNA expression for ATX and LPPs are convincing, as are the functional consequences of these changes. However, its puzzling why corresponding changes in protein levels are not shown, either by western or by fluorescence intensity measurements?

Secondly, results showing special segregation of ATX and SPP1 are convincing, however, given the fact that ATX is secreted, in the context of the tumour micro-environment, can we be sure that cellular polarisation/segregation would have functional consequences? Is it possible that this distribution is just a consequence of hypoxia induced cellular polarisation, which happens in migratory cells? In this context it may be important to know where the LPA receptors are. If they are also in the trailing edge and ATX is secreted, then the mechanistic significance of the special segregation principle is less convincing. In a complex tissue environment, the back of one cell may be in intimate contact with the front of another. However, these are semantic points, depending on available evidence the potential mechanistic significance of the polar segregation needs to be appropriately qualified.

Author Response

POINT #1

Despite the quality of the data presented in this manuscript, there are some surprising gaps in information. For example, do hypoxic changes in ATX/LPP3 mRNA levels lead to corresponding changes in protein expression, or activity? In this context, it is strange that comparative levels of protein expression are not shown under normoxic and hypoxic conditions.  Irrespective of the magnitude of any observed changes, this is an important piece of the jigsaw. If protein levels stay the same, functional modulation may be posttranslational. This is important to know, whatever the result, this should not detract from the value of current data. The magnitude of change imposed by shRNA depletion may not be equivalent to hypoxia-induced changes in gene expression. Therefore, it is important to define the extent to which expression of ATX and LPP proteins changed under hypoxic conditions?

ANSWER

The reviewer is right, some western blots for ATX and LPP3 were already done. We are now providing a new supplementary Figure 1 showing that the increase in ATX and reduction in LPP3 gene expression in hypoxia correlated with respective changes in protein levels.

POINT #2

Equally, as the authors state that the relative localisation of LPP proteins and LPA receptors may affect local LPA concentrations it would be informative to know the relative distribution of LPA receptors under normoxic and hypoxic conditions, are they localised with ATX or LPPs?

ANSWER

This is an excellent point. We are now providing results showing that LPAR1 are uniformly distributed in cells incubated under normoxic or hypoxic condition (new Figure 7A). In addition, the colocalization of LPAR1 with ATX indicates no significant changes in cells incubated under hypoxic condition (new Figure 7B). The new results suggest that while ATX and LPP1 are found polarized in opposite directions, LPAR1 is not segregated with either ATX or LPP1. This is now discussed in the relevant part of the discussion (p13).

POINT #3

Finally, one of the most significant observations in this manuscript is the report that there is a significant reciprocal correlation between the expression of hypoxic genes and LPP3. However, we are not informed if decreased LPP3 expression correlates with stage specific changes or if reduced LPP3 levels correlate with poor prognosis/survival profile etc. This data should be available and would be informative wrt the potential therapeutic significance of these observations.

ANSWER

We agree with the therapeutic importance of these results. We are now providing data indicating that PLPP3 expression can significantly separate low- and high-overall mortality risk groups in sarcoma, glioblastoma and breast cancer patient cohorts (new Figure 2 M-O). Moreover, an increased disparity between low- and high- metastasis-free risk groups was found in sarcoma and breast cancer patients cohorts, the two groups for which metastasis data are available (new Supplemenatary Figure A-B). Interestingly, low PLPP3 gene expression was significantly associated with high-risk groups for both overall survival and metastasis-free survival in all 3 cohorts studied (new supplementary Figure 2 C-G), suggesting that LPP3 hypoxic downregulation is indeed linked to a poor prognosis.

Reviewer 2 Report

“Hypoxia downregulates LPP3 and promotes the spatial segregation of ATX and LPP1 during cancer cell invasion” by Harper et al. reports an interesting phenomenon in which LPA production and degradation are regulated by hypoxia and spatially segregated within cells. A major strength of the manuscript is the use of multiple cell lines for many of the experiments. The paper is very well-written and images and figures are easy to follow and clearly presented. References are thorough and the discussion thoughtful. I have a few points that I believe would help strengthen the conclusions of the manuscript.

Immunoblotting for ATX, LPP1, and LPP3 should be performed in the context of normoxia and hypoxia. ATX knockdown should also be confirmed at the protein level. Hypoxic response should be confirmed with immunoblotting for HIF-1alpha or another marker of hypoxia. It is unclear to what extent the relatively modest differences in mRNA expression would lead to differences in LPA signaling. LPA concentrations should be measured in the context of hypoxia. Are downstream pathways regulated by LPA (Akt, etc.) also affected by hypoxia? Figure 3: Representative images of invadopodia and gelatin degradation should be presented in the main text of supplemental data. In addition, it would be helpful to confirm the existence of invadopodia using multiple immunofluorescence markers (F-actin/phalloidin + Tks5 or cortactin). Scale bars should be provided for all immunofluorescence images.

Author Response

POINT #1

Immunoblotting for ATX, LPP1, and LPP3 should be performed in the context of normoxia and hypoxia.

ANSWER

We are now providing a new supplementary Figure 1 showing that the increase in ATX and reduction in LPP3 gene expression in hypoxia correlated with respective changes in protein levels, while LPP1 levels are not changed.

POINT #2

ATX knockdown should also be confirmed at the protein level.

ANSWER

In the relevant result section of the manuscript (page 7), we now refer to our previous publication indicating that ATX knockdown (using the same cell line and shRNA tool) efficiently down regulates ATX protein levels (both soluble and cell-associated ATX).

POINT #3

Hypoxic response should be confirmed with immunoblotting for HIF-1alpha or another marker of hypoxia.

ANSWER

The hypoxic response of the different cell lines is now confirmed using CAIX, a well-recognized hypoxia responsive gene. Results on CAIX expression are found in panels D-F of the updated Figure 1.

POINT #4

It is unclear to what extent the relatively modest differences in mRNA expression would lead to differences in LPA signaling. LPA concentrations should be measured in the context of hypoxia.

ANSWER

The reviewer is right, only small reduction in LPP3 mRNA expression might not have a significant impact on LPA production/signaling leading to cell invasion. However, and in agreement with the model we are indeed proposing, it is the combination of the diminished expression levels of LPP3 and spatial segregation of ATX and LPP1 at the plasma membrane that would impact LPA production at subcellular locations such as the leading edge of hypoxic cells. In this regard, measurements of total LPA levels in hypoxia might not reflect LPA production/bioactivity at such discrete regions of the plasma membrane.   Also, tools to directly address such a possibility are, to our knowledge, not available yet.

To prevent any misleading, we have modified the abstract and the relevant parts of the introduction to more clearly focus on the overall aim of the manuscript, which is to define whether hypoxia promotes cell invasion by affecting the expression /localization of the main enzymes involved in LPA production (instead of LPA production per se).

POINT #5

Are downstream pathways regulated by LPA (Akt, etc.) also affected by hypoxia?

ANSWER

The answer is yes as we have previously published enhanced PI3K/Akt signaling as effectors of LPAR1/EGFR crosstalk that leads to invadopodia production and cell invasion in hypoxia (Harper et al., Molecular Cancer Research, 2018). Even though LPA was shown to transiently increase Akt phosphorylation under normoxia, a prolonged activation, which was partly LPAR1 driven, was observed in hypoxic conditions. We refer to these results in the introduction section (page 2).

POINT #6

Figure 3: Representative images of invadopodia and gelatin degradation should be presented in the main text or supplemental data.

In addition, it would be helpful to confirm the existence of invadopodia using multiple immunofluorescence markers (F-actin/phalloidin + Tks5 or cortactin).

ANSWER

-Representative images of gelatin degradation are now presented in a new Figure 3 F.

-We have previously published that hypoxia, and LPA through LPA1 induce actin/cortactin-rich invadopodia protrusions in the HT-1080 cell line (Harper et al., Cancer Res, 2010; Arsenault et al., PLOSone, 2013; Harper et al., Molecular Cancer Res, 2018). We now refer to these results in the relevant result section (page 4).

POINT #7

Scale bars should be provided for all immunofluorescence images.

ANSWER

Scale bars are now provided for all microscopy images.

Round 2

Reviewer 2 Report

The authors have thoughtfully and completely addressed all of my previous comments.